# Molecular Docking and Dynamics Simulation of Protein β-Tubulin and Antifungal Cyclic Lipopeptides

**DOI:** 10.3390/molecules24183387

**Published:** 2019-09-18

**Authors:** Nubia Noemi Cob-Calan, Luz America Chi-Uluac, Filiberto Ortiz-Chi, Daniel Cerqueda-García, Gabriel Navarrete-Vázquez, Esaú Ruiz-Sánchez, Emanuel Hernández-Núñez

**Affiliations:** 1Tecnológico Nacional de Mexico, Instituto Tecnológico de Conkal, Conkal C.P.97345, Yucatán, Mexico; nubia.cob@itconkal.edu.mx; 2Departamento de Física Aplicada, CINVESTAV-IPN Unidad Mérida, Mérida C.P. 97310, Yucatán, Mexico; chiamerika@hotmail.com; 3CONACYT-Universidad Juárez Autónoma de Tabasco, Centro de Investigación de Ciencia y Tecnología Aplicada de Tabasco, Cunduacán C.P.86690, Tabasco, Mexico; filiberto.ortiz@ujat.mx; 4CONACYT-Departamento de Recursos del Mar, CINVESTAV-IPN Unidad Mérida, Mérida C.P.97310, Yucatán, Mexico; dacegabiol@ciencias.unam.mx; 5Facultad de Farmacia, Universidad Autónoma del Estado de Morelos, Cuernavaca C.P.62209, Morelos, Mexico; gabriel_navarrete@uaem.mx

**Keywords:** molecular dynamics, molecular docking, β-tubulin, cyclic lipopeptides, antifungal activity

## Abstract

To elucidate interactions between the antifungal cyclic lipopeptides iturin A, fengycin, and surfactin produced by *Bacillus* bacteria and the microtubular protein β-tubulin in plant pathogenic fungi (*Fusarium oxysporum*, *Colletrotrichum gloeosporioides*, *Alternaria alternata*, and *Fusarium solani*) in molecular docking and molecular dynamics simulations, we retrieved the structure of tubulin co-crystallized with taxol from the Protein Data Bank (PDB) (ID: 1JFF) and the structure of the cyclic lipopeptides from PubChem (Compound CID: 102287549, 100977820, 10129764). Similarity and homology analyses of the retrieved β-tubulin structure with those of the fungi showed that the conserved domains shared 84% similarity, and the root mean square deviation (RMSD) was less than 2 Å. In the molecular docking studies, within the binding pocket, residues Pro274, Thr276, and Glu27 of β-tubulin were responsible for the interaction with the cyclic lipopeptides. In the molecular dynamics analysis, two groups of ligands were formed based on the number of poses analyzed with respect to the RMSD. Group 1 was made up of 10, 100, and 500 poses with distances 0.080 to 0.092 nm and RMSDs of 0.10 to 0.15 nm. For group 2, consisting of 1000 poses, the initial and final distance was 0.1 nm and the RMSDs were in the range of 0.10 to 0.30 nm. These results suggest that iturin A and fengycin bind with higher affinity than surfactin to β-tubulin. These two lipopeptides may be used as lead compounds to develop new antifungal agents or employed directly as biorational products to control plant pathogenic fungi.

## 1. Introduction

Lipopeptides are a class of proteins with antibiotic properties against a wide range of microorganisms, including plant pathogenic fungi [1]. They consist of a lipid tail linked to a short oligopeptide that can be linear or cyclic [2]. The ring of the cyclic lipopeptides is formed by an ester (lactone) or amide (lactam) group. Cyclic lipopeptides are produced as secondary metabolites by a variety of bacterial genera, including *Actinomycetes*, *Streptomyces*, *Bacillus*, and *Pseudomonas* [2,3,4].

Cyclic lipopeptides produced by *Bacillus* are classified into three categories: iturins, fengycins, and surfactins [1,2,5]. They are all fungicidal against various species of plant pathogenic fungi within the groups Deuteromycota, Basidiomycota, and Ascomycota [6,7,8]. The antifungal action includes the damage and dysfunction of membranes and organelles, alterations in mitochondrial membrane potential, oxidative stress, and chromatin condensation [8]. Although the membrane damage caused by permeabilization and ion-conducting pore formation in the cell membrane has been characterized in detail [9,10], other target sites of these compounds have been scarcely studied. In this context, the β-tubulin protein may be a suitable target site, because it has been highly implicated as the target of a wide range of synthetic fungicides [11,12]. Fungicides act on β-tubulin by inhibiting the assembly of heterodimers of α- and β-tubulin into microtubules, which are vital to various cellular processes such as cell signaling, cell motility, cell division, and mitosis [13]. β-tubulin-targeting agents may bind to the protein at any of four sites: laulimalide, taxane/epothilone, vinca alkaloid, and colchicine sites [11,12]. These binding pockets interact with active compounds based on the hydrophobicity and size of the molecule. Two binding pockets, taxane/epothilone and vinca alkaloid, interact with large and complex molecules, opening new avenues to explore large compounds as potential β-tubulin target agents [11,12]. Compounds that bind taxane/epothilone are structurally and chemically similar to the cyclic lipopeptides in terms of the number of carbons in the ring [14,15,16].

The search for new candidate target molecules, including β-tubulin, in pathogenic fungi and other pathogens of plants has become more efficient with the aid of computer-simulated molecular docking studies. For example, in the fungus *Fusarium oxysporum*, 2,4-di-*tert*-butylphenol, a compound that effectively inhibits spore germination and hyphal growth in vitro, was discovered through molecular docking analyses to potentially bind to β-tubulin protein at the HIS 118 and THR 117 residues [17]. In the plant parasitic nematode *Bursaphelenchus xylophilus*, the compounds amocarzine, mebendazole, and flubendazole were revealed to bind various target sites, including β-tubulin, by analyzing hypothetical binding sites in a protein model constructed through molecular docking [18].

Here, we used in silico studies of the molecular interaction of the cyclic lipopeptides iturin A, fengycin or surfactin with β-tubulin to identify which molecules have the highest affinity for binding β-tubulin from four plant pathogenic fungi. Modified β-tubulin was stabilized and validated. Coupling studies of the cyclic lipopeptides showed that all three can potentially bind to β-tubulin; that iturin A and fengycin bind with higher affinity than surfactin and are the most promising antifungal candidates.

## 2. Results and Discussion

### 2.1. Homology Modeling of β-Tubulin

The search for homologs of the β-tubulin template using the protein Basic Local Alignment Search Tool (blastp) yielded positive hits with 84% similarity for a peptide sequence from four species of fungi: *F. oxysporum*, *Colletrotrichum gloeosporioides*, *Alternaria alternata*, and *Fusarium solani* obtained from PubChem. Analysis of these sequences with HMMER software identified conserved domains of the protein families of tubulin (FtsZ amino acids 3–212) and tubulin C (amino acids 261–382) with an e-value < 7e^−42^ (Figure 1 and Appendix A). Hidden Markov models implemented in the HMMER3 software are robust and have been used for remote protein homology detection [19]; thus, with the results of the HMMER analysis, we confirmed that the template protein and the proteins of the four species of fungi are homologous. In addition, we also found that the catalytic site (where taxol binds as a ligand) was constant in the protein β-tubulin of the four fungal species analyzed. In a similar study, De Pereda et al. [20] found similarity among several tubulins, with the following proportions: 33% for α, 21% for β, and 45% for other secondary structures. Carpenter et al. [21] evaluated 475 tubulin isotypes, finding global and structural similarities, but differences among net electric charges, solvent accessible surface areas, and electrical dipole moments. They concluded that these differences are responsible for the variations in their cellular activity. In a blastp search using tubulin from the nematode *Bursaphelenchus xylophilus* as the query, the X-ray crystal structure of ID 5IJ0 in the Protein Data Bank (PDB) was identified as homologous and shared 86% similarity with the nematode tubulin [18].

### 2.2. Active Site Validation

The active site of β-tubulin was validated with the co-crystallized native ligand taxol. Comparison of the poses obtained by of the AutoDock Vina program against those of the crystallized protein yielded root mean square deviation (RMSD) = 1.56 Å, indicating an appropriate optimization score [22,23]. These values are small and support binding at the simulation site with the original orientation of the co-crystallized molecule (Figure 2).

### 2.3. Molecular Docking Analysis

In the analyses to explore the binding affinities of the three-antifungal cyclic lipopeptides with the potential target site in β-tubulin, iturin A, fengycin, and surfactin exhibited docking interactions with β-tubulin. Figure 3 shows the optimized structures of these compounds, using a stochastic global conformational search strategy interfaced to the Gaussian 16 code [24].

Docking of the alkaloid taxol, used as the native ligand, with the modeled β-tubulin protein was simulated using the AutoDock Vina open source program and a genetic algorithm. To explore the binding affinity and the molecular basis of the interactions, the three lipopeptides were docked within the binding sites predicted from the β-tubulin model. The evolutionary metaheuristic used showed that iturin A, fengycin, and surfactin exhibited docking poses with high binding affinities (in terms of coupling energy). The binding energies and amino acid interactions for each compound with β-tubulin are given in Table 1. The docking analysis of the compounds with β-tubulin generated negative values for free energy, suggesting high affinity for the binding pocket. Although all the binding conformations of each compound in the active binding pocket involved both H-bond and non-bonded interactions, iturin A and fengycin had higher binding affinities for β-tubulin based on lower binding energies.

The binding modes of iturin A, fengycin, and surfactin within the binding site of β-tubulin are important for the design of highly toxic compounds against plant pathogenic fungi. The binding interactions of these compounds with the active site are stabilized through H-bonds and non-bonded interactions (Appendix A). Generally, a non-bonded interaction (e.g., van der Waal interaction) contributes to a more stable protein–ligand complex and thus greater antimicrobial activity [26]. From the docking results, the binding energy for β-tubulin with iturin A (−7.0 kcal/mol) is equal to that for β-tubulin with fengycin (−7.0 kcal/mol). The binding energy for β-tubulin with surfactin was −6.3 kcal/mol (Figure 4). The *Ki* value for the binding of taxol with β-tubulin was lower (*Ki* = 0.20 μM) compared to those observed for the binding of iturin A or fengycin with β-tubulin (*Ki* = 7.084 μM for each). The *Ki* value for the binding of surfactin with β-tubulin was high (*Ki* = 23.18 μM), indicating that this cyclic lipopeptide had the lowest affinity of the three (Table 1).

The docking analysis revealed that iturin A and fengycin have equal binding affinity and higher than surfactin for β-tubulin based on the docking energy and number of H-bonds. Iturin A binds to β-tubulin with two H-bonds, and to fengycin with three (Table 1). For iturin A, the amino acid in position 68 forms an H-bond with the oxygen of Pro274 of β-tubulin with a bond length of 2.38 Å. Similarly, another hydrogen bond forms between the amino acid in position 66 of iturin A and the oxygen of Thr276 of β-tubulin with a bond length of 2.16 Å. For fengycin, the oxygen in position 114 forms an H-bond with the oxygen of Glu27 and Thr276 of β-tubulin with a bond length of 2.11 Å. The second and third H-bonds are formed between the nitrogen at position 115 of the fengycin, which connects with the oxygen and carbonyl of the Thr276 of β-tubulin (bond lengths of 2.43 and 2.09 Å, respectively). All details of the atoms involved in bonding with ligands, bond lengths, docking energies, and *Ki* values are given in Table 1.

Hypothetically, the higher the affinity of a ligand for its target protein, the more effective its activity at the cellular or organism level. In that case, iturin A and fegycin would be more toxic than surfactin. In agreement with this assumption, experiments with plant pathogenic fungi showed that iturin A and fengycin were more effective than surfactin at inhibiting mycelial growth and reducing the severity of leaf damage [27,28].

Again, in a hypothetical interaction between iturin A or fengycin with β-tubulin in plant pathogenic fungi, cellular functions such as spore germination and hyphal growth needed for pathogenesis would be impaired due to the inhibition of microtubule formation or an alteration in microtubule stability [11]. Iturin A and fengycin may thus have great potential as biorational fungicides. However, further evaluation of the effectiveness of these lipopeptides for disease control is required.

It is important to note that, based on the results of the present study, we cannot establish the specific mechanism of action of the lipopeptides on β-tubulin or classify them as either microtubule stabilizers or destabilizers. Further studies are necessary to assess their impact on dimer stability and their toxicity.

### 2.4. Molecular Dynamics (MD)

To assess the stability of the β-tubulin–iturin A, β-tubulin–fengycin, and β-tubulin–surfactin complexes, we used the following descriptors: ligand RMSD, interface RMSD (IRMSD), number of hydrogen bonds, and distance between the center of mass of the binding site and the ligand.

The ligand RMSD is displayed in Figure 5a. The ligand backbone RMSD was estimated using the structure in the equilibrated stage of the simulation (frame at 50 ns) as a reference. The ligand RMSDs remained below 2.5 Å, with the fengycin backbone RMSD being the smallest and the most stable (smallest deviation in the RMSD).

The interface RMSD (IRMSD) is shown in Figure 5b. Because the IRMSD includes all the amino acids in the protein that are closest to the ligand (less than 4 Å from the ligand), it can give insight into the stability of the interacting amino acids. All IRMSDs were below 3 Å, while surfactin and iturin A had the lowest IRMSD values, indicating the highest stability.

The number of H-bonds is given in Figure 5c. Iturin A and fengycin had the most H-bonds, but the fengycin H-bonds were more stable over the entire simulation and would likely play a significant role in stabilizing the protein–ligand interactions. The H-bonds in docking structures were maintained during the Molecular Dynamics (MD) simulations; in addition, other H-bonds were observed.

The analysis of the distance between the center of the mass of the protein and the center of the mass of each ligand showed that the shortest distance was found for iturin A (Figure 5d).

The protein–ligand complexes formed are thermally stable at 300 K. Ligands do not tend to detach over 100 ns of simulation in a realistic solvent environment, depending on the distance, the number of H-bonds, and the IRMSD and ligand RMSD analyses. We conclude that iturin A and fengycin were more stable over the simulation.

## 3. Materials and Methods

### 3.1. Domain Identification and Template Search

The sequence of the crystalized β-tubulin protein was downloaded from the Protein Data Bank (ID:1JFF was used as a template) and used as the search query in the online platform of blastp [29] to find homologous protein sequences for the phytopathogenic fungi *Fusarium oxysporum* (XP_018241948.1), *Colletrotrichum gloeosporioides* (XP_ELA34262.1), *Alternaria alternata* (XP_018387617.1), and *Fusarium solani* (XP_003046201.1). The homologous proteins that were identified by significant hits (similarity > 70%, e-value < 1e^−10^) were then used in a search against the Pfam database for conserved domains of the β-tubulin family using the software HMMER v3.2.1.

### 3.2. Conformational Search of the Lipopeptides

Initial structures for iturin A (CID:102287547), fengycin (CID:100977820), and surfactin (CID:10129764) were obtained from the PubChem database (https://pubchem.ncbi.nlm.nih.gov). These molecules have a high number of degrees of torsional freedom, so the global conformational search for their rotational isomers was performed in two stages. In the first stage, a stochastic algorithm generated 10^4^ rotational isomers, all dissimilar to each other. All the generated structures were optimized at the semi-empirical level PM6-D3 [30,31], as implemented in the MOPAC code [32]. In the second stage, the best 10^3^ energetically favorable rotational isomers were chosen to be fully optimized at the Density Functional Theory (DFT) level through the Gaussian 16 code [24]. The GGA hybrid functional B3LYP [33] was chosen in conjunction with the 6-31G(d) basis set [34], usually used for organic molecules.

### 3.3. Validation of Active Site

The active site on β-tubulin was validated using taxol as the native ligand (PDB: 1JFF). RMSD was set to less than 2 Å to determine the best docking position between β-tubulin and the ligands using Autodock Vina [35]. The validation was performed with 10 poses, each in triplicate, selecting the lowest energy value. Protein visualization and overlap were carried out using Pymol 3.1 (Schrödinger, San Diego; http://www.pymol.org/).

### 3.4. Molecular Docking

The docking of β-tubulin with each ligand (iturin A, fengycin, and surfactin) was simulated using the program AutoDock Vina, which has been used to estimate the conformation of protein–ligand complexes [35] and significantly improves the average accuracy of the binding mode predictions.

All calculations for protein-fixed ligand-flexible docking were analyzed using the Lamarckian Genetic Algorithm (LGA) method [22]. The docking site on β-tubulin was defined by establishing a grid box using Pymol 3.1 (Appendix A). The grid box size for the coordinates *x*, *y*, and *z* was 30 Å, with a grid spacing of 0.375 Å, centered on *x* = −2.265, *y* = 7.164, and *z* = 17.568 Å. The best conformation was chosen based on the lowest binding energy after the docking search was completed. Autodock Vina was set for 1000 modes and exhaustiveness 1000 (exhaustiveness of the global search, roughly proportional to time) for each ligand structure, and for each run, the best pose was saved. The average binding energy for the best poses was used as the final binding energy value. This process was repeated three times.

### 3.5. Molecular Dynamics

For each protein–ligand system, four poses with minimum energy were evaluated (10, 100, 500, 1000) with three replicas each (R1, R2, R3). Systems were previously minimized; the number of particles, volume, and temperature (NVT) as well as the number of particles, pressure, and temperature (NPT) were equilibrated; then, MD simulations in water over 100 ns were run.

All calculations (minimization, NVT equilibration, NPT equilibration, and MD simulations) were performed using GROMACS 5.1.2 [36] with GPU support. We used the GROMOS96 43A1 force field for the protein and the ligand in all cases. We used the simple point-charge (SPC)model for water molecules. The molecules were solvated in a dodecahedron box, with periodic boundary conditions and a minimum distance between the solute and the box of 1.5 nm. Sodium and chloride ions were added to neutralize the system. Energy minimization steps were carried out using the steepest descent algorithm. During NVT ensemble simulations, harmonic position restraints were applied to the solute heavy atoms with a force constant of 1000 kJ mol^−1^ nm^−2^. The MD production runs were performed using 2 fs as the time step. For pressure coupling, we used Berendsen at 1 bar. Temperature was controlled by setting the Langevin dynamics at 300 K. Standard *xyz* periodic boundary conditions were considered. A cut-off distance of 1.0 nm for the Coulomb and van der Waals neighbor list was updated according to the Verlet cut-off scheme. The long-range part of the Coulomb interactions was evaluated using the particle-mesh Ewald (PME) method with a relative tolerance of 5–10, order of 6, and Fourier spacing of 0.1. All bonds were constrained using linear constraint solver for molecular simulations (LINCS), while SETTLE was used for constraining the water molecules.

For the analysis of MD simulations, we assessed the ligand root mean square deviation (RMSD), interface root means square deviation (IRMSD), number of H-bonds, and distances between the protein-binding sites and the ligand. The ligand RMSD was calculated for the backbone using an equilibrated frame at 50 ns as a reference. For the IRMSD, we first aligned the backbone protein trajectory to the backbone of the crystal structure. Then, we grouped clusters by similar conformations of the MD trajectory and chose the middle structure of the largest cluster as a representative element. We then found the closest amino acids to the ligand (less than 4 Å) of these representative structures and calculated the RMSD of those amino acids over the complete 100 ns MD simulation. We also calculated the distances between the backbone center of mass of the protein-binding site and the backbone center of mass of each ligand. We considered the first 50 ns of the simulation as an equilibration period, after which we presented the results over the last 50 ns of MD simulation.

## 4. Conclusions

In conclusion, using molecular docking and molecular dynamics simulation, we demonstrated the interactions of the cyclic lipopeptides iturin A, fengycin, and surfactin with β-tubulin. The analysis of molecular docking showed that residues Pro274, Thr276, and Glu27 of β-tubulin are responsible for the formation of hydrogen bonds. Iturin A and fengycin had higher binding affinity than surfactin to the catalytic site of β-tubulin. These molecules can potentially target β-tubulin in plant pathogenic fungi. These cyclic lipopeptides can either be tested directly as biorational fungicides or used as lead compounds, thus opening new avenues to biorationally design antifungal agents.

## Figures and Tables

**Figure 1 molecules-24-03387-f001:**
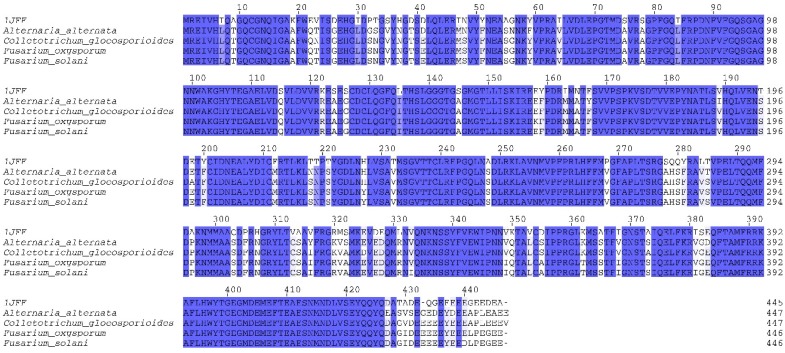
Alignment of the peptide sequence from the four fungal species with the template sequence of β-tubulin (1JFF) obtained from the PubChem. Highly conserved regions are indicated in blue.

**Figure 2 molecules-24-03387-f002:**
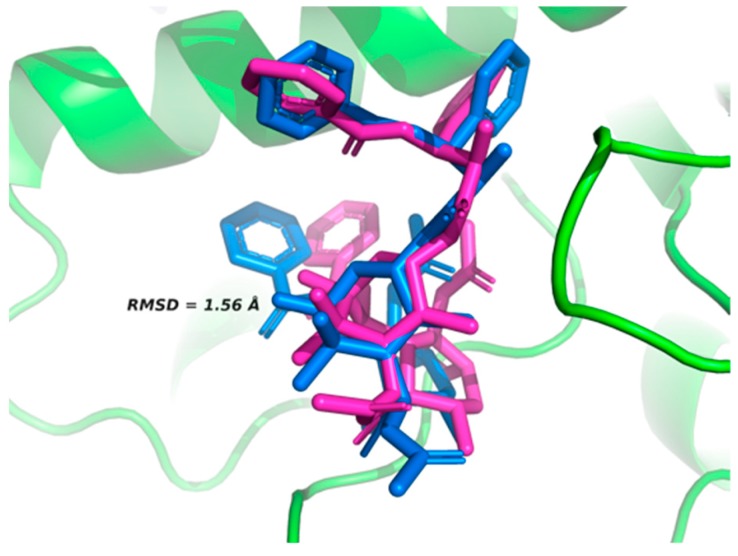
Ligand-binding site of β-tubulin protein with co-crystalized native taxol (blue) and taxol as posed by the Autodock Vina program (pink).

**Figure 3 molecules-24-03387-f003:**
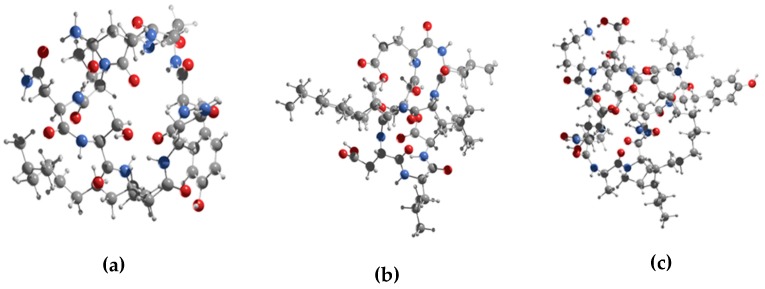
Three-dimensional structural representation of (**a**) iturin A, (**b**) fengycin, and (**c**) surfactin calculated by a homemade code interfaced to Persistence of Vision Ray-tracer (POVRAY.)

**Figure 4 molecules-24-03387-f004:**
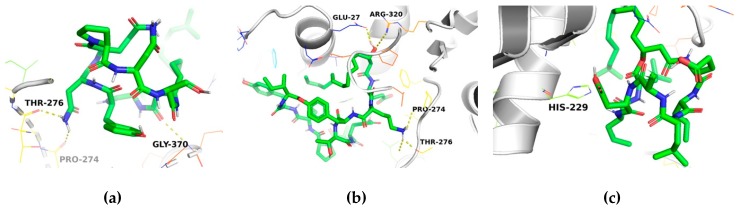
Molecular docking simulation showing the interaction of (**a**) iturin A, (**b**) fengycin, and (**c**) surfactin (green and blue) with active site residues of β-tubulin (grey).

**Figure 5 molecules-24-03387-f005:**
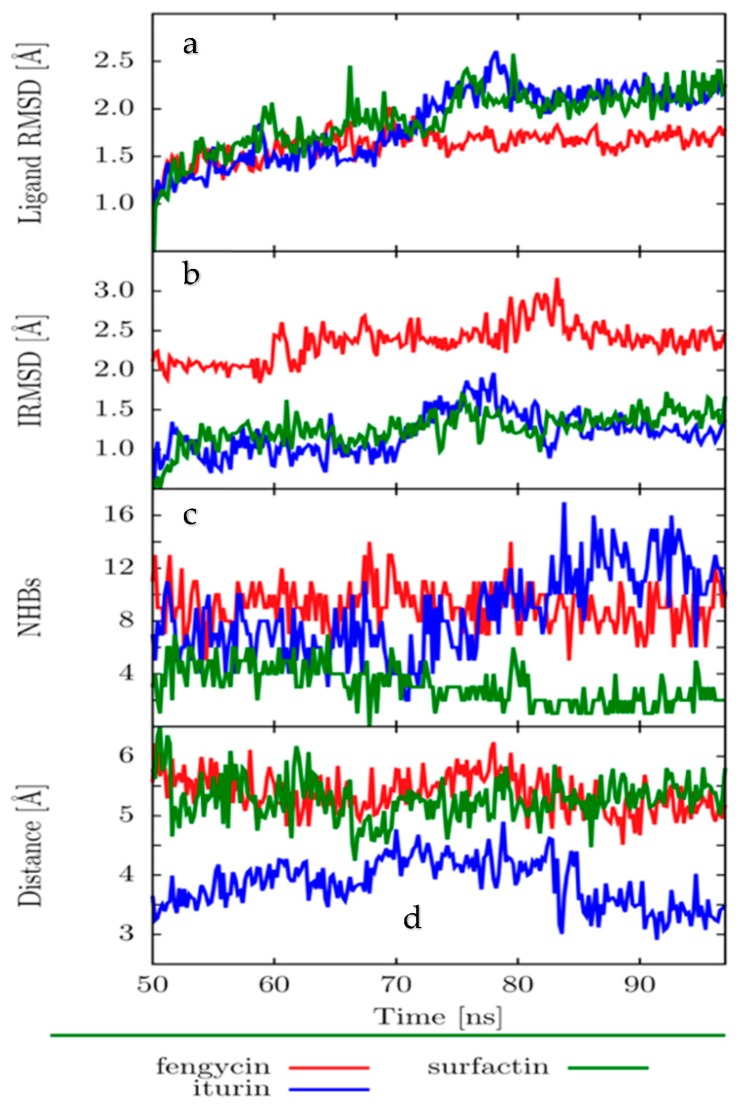
Time evolution of (**a**) the ligand root means square deviation (RMSD), (**b**) the interface RMSD (IRMSD), (**c**) the number of H-bonds between the protein and ligand, and (**d**) the distance between the ligand and tubulin-binding site. Analysis was performed by Gromacs.

**Table 1 molecules-24-03387-t001:** Binding affinity for the molecular coupling in the β-tubulin protein complex with taxol and the test lipopeptides.

Compound	No. of H-Bonds	Residue Receptor	Ligand	Bond Length (Å)	Docking Score (kcal/mol)	*k_i_* (μM^) a^
Taxol	1	Thr276(N)	O06	2.92	−9.1	0.202
Iturin A	2	Pro274(O) Thr276(O)	(ND2)68 (ND2)66	2.38 2.16	−7.0	7.084
Fengycin	3	Glu27(OE2) Thr276(O) Thr276(OG1)	(O)114 (N)115 (N)115	2.11 2.43 2.09	−7.0	7.084
Surfactin	1	His229(NE2)	H	2.7	−6.3	23.188

**^a^***Ki =* e^-^^ΔG/RT^
*ΔG* = Gibbs free energy; *R* = 1.9872 cal/mol.K; *T* = 298.15 °K [25].

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
