# Peer review of "Molecular Docking and Dynamics Simulation of Protein β-Tubulin and Antifungal Cyclic Lipopeptides"

_molecules, 2019, doi:10.3390/molecules24183387_

Round 1
Reviewer 1 Report
In this paper, the authors used some common computational chemistry methods, including Homology modeling, Molecular docking and molecular dynamics to study the interactions of cyclic lipopeptides (iturin A, fengycin, and surfactin) and tubulin from phytopathogen fungus (Fusarium oxysporum, Colletrotrichum gloeosporioides, Alternaria alternata y Fusarium solani). The results suggest that iturin A and fengycin have higher activity than surfactin against phytopathogen fungi. The methods used are common, and the results seem logical, may provide some insights into the interactions between protein β-tubulin and cyclic lipopeptides, as well as further drug design.
However, I do not think that this work can give new insights into this issue of “Computational Enzymology”. This work may be more suitable for some special computational chemistry Journal, such as Computational and Theoretical Chemistry
https://www.journals.elsevier.com/computational-and-theoretical-chemistry/
Author Response
We thank the reviewer comments.
We believe that the manuscript fits within the section “Computational and Theoretical Chemistry” of Molecules, where the focus of the section is the application and development of computational and theoretical chemistry. Our intention when submitting the manuscript to Molecules was to reach the audience of the journal as done by other high-quality papers published in Molecules that are similar to our work in terms of focus of the study and methodologies.
The manuscript was submitted to the special issue on Computational Enzymology as the keywords for this issue included “Inhibition” and “Molecular dynamics”. However, we believe that the manuscript may also be suitable for the special issue on Recent Advances in Computational Drug Discovery: From in Silico Screening to Multiscale De Novo Drug Design, where the keywords for this issue includes virtual screening, cheminformatics and bioinformatics.

Reviewer 2 Report
The article titled Molecular docking and molecular dynamics simulation of protein β-tubulin and cyclic lipopeptides encompasses computational study of interactions of cyclic lipopeptides with β-tubulin. In my opinion, it is an interesting area of research, however, this article needs serious attention and inputs from the authors to make the case more convincing.
My major comments and suggestions are as follows:
It is little confusing why authors chose these three cyclic lipopeptides towards binding against β-tubulin. Authors needs to clarify and give more inputs in the background section in order to justify the reason behind studying these compounds as a potential inhibitor of β-tubulin. Many places in the article has grammatical errors and spelling mistakes, so authors need to consider rewriting those sections. For example, the pdb code for the β-tubulin-taxol complex was mentioned as IJFF as supposed to 1JFF. Line number 145 to 169 needs to be rewritten. In line number 151 and 152, authors mentioned “Thr D276, GlyE370, AspC26, AspD226, ArgE369, HisD229, ArgD278, LeuD275, 152 PheD272, ProD274, LeuD217, LeuD219, ProD360 and LeuE371”, could not get what is the D, C and E stands for. In line number 136 to 140 authors mentioned about Ki values of the cyclic lipopeptides, it is unclear how this value was calculated. If it was adapted from any previous work, then that work needs to be cited. In line number 164 to 166 authors mentioned “docking results also supported that compound 2,4-di-ter-butylphenol is a potent inhibitor of F. oxysporum. The current study provide useful information regarding the antifungal activity of the bioactive compounds against the plant pathogenic fungi F. oxysporum from P. monteilii PsF84” The reason behind this statement is unclear. This article claims that binding of cyclic lipopeptides destabilizes microtubule association. In order to assess that authors could consider simulating tubulin heterodimer complexed with or without cyclic lipopeptides to see if it has any impact on the dimer stability. Authors should consider expanding and rephrasing conclusion part.
Author Response
Comment - general. The article titled Molecular docking and molecular dynamics simulation of protein β-tubulin and cyclic lipopeptides encompasses computational study of interactions of cyclic lipopeptides with β-tubulin. In my opinion, it is an interesting area of research, however, this article needs serious attention and inputs from the authors to make the case more convincing.
Response. We thank the reviewer comment. We have made all changes suggested throughout the manuscript. To build the new version of the manuscript we requested the service of a professional (Willows End Scientific Editing and Writing / Beth E. Hazen) to correct style, grammar and structure.
Comment 1. It is little confusing why authors chose these three cyclic lipopeptides towards binding against β-tubulin. Authors needs to clarify and give more inputs in the background section in order to justify the reason behind studying these compounds as a potential inhibitor of β-tubulin.
Response. We have taken the reviewer comment. We have extended the background information related the characteristics of the β-tubulin binding sites in the Introduction. The paragraph gives rationale behind studying these compounds as potential inhibitors β-tubulin. Lines 54-62.
Comment 2. Many places in the article has grammatical errors and spelling mistakes, so authors need to consider rewriting those sections.
Response. The new version of the manuscript was revised by a professional from Willows End Scientific Editing and Writing. Style, grammar and structure have been corrected.
Comment 3. The pdb code for the β-tubulin-taxol complex was mentioned as IJFF as supposed to 1JFF.
Response. This has been corrected. The code IJFF has been changed to 1JFF. Lines 24 and 97, and Figure 1.
Comment 4. Line number 145 to 169 needs to be rewritten.
Response. We have taken the reviewer comments. The paragraph has been rewritten. Emphasis has been placed now in the description of binding sites between lipopeptides and β-tubulin (Lines 145-154), as well as in the discussion of supporting assumptions from molecular docking studies based on in vitro assays (Lines 155-164).
Comment 5. In line number 151 and 152, authors mentioned “Thr D276, GlyE370, AspC26, AspD226, ArgE369, HisD229, ArgD278, LeuD275, 152 PheD272, ProD274, LeuD217, LeuD219, ProD360 and LeuE371”, could not get what is the D, C and E stands for.
Response. We have corrected all typographical errors. Lines 145-154.
Comment 6. In line number 136 to 140 authors mentioned about Ki values of the cyclic lipopeptides, it is unclear how this value was calculated. If it was adapted from any previous work, then that work needs to be cited.
Response. We included in the new version of the manuscript; the formula used to calculate Ki (Line 129). Ki was calculated by the formula: , where Ki = Inhibition constant; –G = Gibbs free energy; T = temperature, 298.15°K; and R = gas constant, 1.9872 cal/mol.K. This formula has been used in previous studies by González-Trujano et al. Biomed. Pharmacotherapy. 2018, 101: 553–562. The reference has been cited in Line 353.
Comment 7. In line number 164 to 166 authors mentioned “docking results also supported that compound 2,4-di-ter-butylphenol is a potent inhibitor of F. oxysporum. The current study provides useful information regarding the antifungal activity of the bioactive compounds against the plant pathogenic fungi F. oxysporum from P. monteilii PsF84” The reason behind this statement is unclear.
Response. We have restructured the paragraph (Lines 155-164). We have considered that the statement was out of context. This has been deleted.
Comment 8. This article claims that binding of cyclic lipopeptides destabilizes microtubule association. In order to assess that authors could consider simulating tubulin heterodimer complexed with or without cyclic lipopeptides to see if it has any impact on the dimer stability.
Response. We have corrected the sense of the paragraph (lines 167-170). With the results obtained in the present study, it is difficult to infer the specific action of the lipopeptides on microtubule assembly. We have stated that further studies are necessary to find out the role of the lipopeptides on the dimer stability (Lines 165-167).
Comment 9. Authors should consider expanding and rephrasing conclusion part.
Response. We have taken the reviewer comment, the Conclusion section has been rephrased and restructured. Lines 268-274.

Round 2
Reviewer 1 Report
Yes, I have seen a lot of changes in the revised version.The scientific content of this paper is OK. I think that the guest edit is a good judgment if this work can be publicised in this special issue.
Reviewer 2 Report
Looks good now!